# Mind the Gap! Core-Peripheral Temperature Gradient and Its Relationship to Mortality in Major Burns

**DOI:** 10.3390/ebj6010011

**Published:** 2025-03-02

**Authors:** Niamh Keohane, Jennifer Driver, Randeep Mullhi, Elizabeth Chipp, Barbara Torlinska, Tomasz Torlinski

**Affiliations:** 1Worcestershire Royal Hospital, Worcestershire Acute Hospitals NHS Trust, Worcester WR5 1DD, UK; niamh.keohane3@nhs.net; 2Southmead Hospital, North Bristol NHS Trust, Bristol BS10 5NB, UK; jmjdriver@gmail.com; 3West Midlands Burn Centre, Queen Elizabeth Hospital, University Hospitals Birmingham NHS FT, Birmingham B15 2GW, UK; randeep.mullhi@uhb.nhs.uk (R.M.); elizabeth.chipp@uhb.nhs.uk (E.C.); 4Department of Applied Health Sciences, College of Medicine and Health, University of Birmingham, Birmingham B15 2TT, UK; 5Aston Medical School, College of Health and Life Science, Aston University, Birmingham B4 7ET, UK

**Keywords:** burn, resuscitation, body temperature, core-peripheral gap, mortality, critical care, intensive care, thermoregulation, core temperature, thermal injury

## Abstract

The association between hypothermia and poor outcomes in severe burn injury is well established. However, the significance of the core-peripheral temperature gradient has not previously been investigated. Institutional guidance at our burns centre advocates avoiding hypothermia and targeting a body temperature between 37.5 and 39.5 °C. The core-peripheral temperature gap should be ≤2 °C, based on expert opinion. Data from 61 patients admitted to the Intensive Care Unit (ICU) with severe burns between 2016 and 2022 were analysed. A higher core temperature at 48 h, avoidance of hypothermia and a core-peripheral temperature gap > 2 °C were associated with reduced odds of mortality. The mean core body temperature and core-peripheral temperature gap increased over the first 48 h (r = 0.5, *p* < 0.001). All non-survivors had a core-peripheral gap < 2 °C at 48 h. Survivors had a higher mean 48 h gap (1.6 [95%CI:1.3–1.9]) than non-survivors (0.8 [95%CI:0.2–1.4; *p* = 0.04]). Our findings support previous studies suggesting that avoiding hypothermia and achieving a higher target temperature are associated with reduced mortality. However, it challenges the previous expert consensus that a lower core-peripheral gap indicates better outcomes. Further research with a larger cohort of patients is required to identify whether a higher core-peripheral temperature gap predicts outcomes in critically ill patients with severe burns.

## 1. Introduction

The importance of the avoidance of hypothermia in patients with burn injuries is well documented. The International Society for Burn Injuries (ISBI) recommends the avoidance of hypothermia, defined as a core temperature of <36 °C [1]. Local guidelines at the Regional Burns Centre at University Hospitals Birmingham NHS Foundation Trust, United Kingdom, advocate avoiding hypothermia as per the ISBI guidelines. Additionally, they recommend the maintenance of a core-peripheral temperature gap of ≤2 °C in the initial 48 h following a severe burn injury, based on local experts’ consensus. To our knowledge, the use of core-peripheral temperature gradients as a predictor of prognosis in critically ill patients with major burn injuries has not previously been investigated.

Thermoregulation remains a significant challenge in critically ill burn patients. The early onset of hypothermia is associated with increased mortality and length of hospital stay [2]. This particular challenge is compounded by varied staff perceptions and practices in monitoring temperature, temperature targets and interventions to achieve these targets. Such variation has been highlighted by recent surveys in North America and the UK [3,4]. These surveys identified that although the measurement of core body temperature is standard practice, the measurement of peripheral temperature is much more variable. Of the UK burns centres surveyed, 67% routinely measured core and peripheral temperatures in an Intensive Care Unit (ICU) setting. Some respondents measured core and peripheral temperatures together in the initial 48–72 h following injury but, after this time, measured core temperature in isolation [3].

The core-peripheral temperature gradient is the temperature difference between the core body temperature and the temperature of the peripheries (i.e., skin temperature). In healthy individuals, warm peripheries indicate good perfusion; however, the gradient exists due to loss of heat via radiation [5]. This loss is minimised via counter-current heat exchange for energy efficiency and a core-peripheral temperature gradient in a healthy individual is usually maintained around 2–4 °C [5].

In a diseased state, a raised core-peripheral temperature gradient is established as a non-invasive indicator of shock and occurs as a result of reduced effective circulating volume and heightened sympathetic nervous system activation, triggering peripheral vasoconstriction to prioritise perfusion of vital organs [6]. However, the quantitative value of the core-peripheral temperature difference is still controversial. Isben first discovered the relationship between peripheral temperature and cardiac function in the 1960s [7]. This finding was supported by work by Joly and Weil, who conducted a study on 100 patients with signs of circulatory shock and found a statistically significant correlation between worsening cardiac index and decreased toe temperature [8]. In recent years, Amson and colleagues conducted a prospective observational study to determine the association between core-to-skin temperature gradient and day 8 mortality in patients with septic shock. Their findings showed that a temperature gradient > 7 degrees predicted 8-day mortality, independently of the Sequential Organ Failure Assessment (SOFA) score. The core-peripheral temperature gradient correlated with other markers of hypoperfusion, including capillary refill time, mottling score and lactate level [9].

Contrary to these findings, an older study by Woods and colleagues demonstrated no association between temperature gradient and invasive measures of circulating blood volume, including cardiac index, systemic vascular resistance and systemic vascular resistance index [10]. Many research groups have concluded that although a high core-peripheral temperature gradient may be a helpful indicator of shock, it should only be used as an adjunct to other clinical signs and haemodynamic variables [11,12]. When used in isolation, a core-peripheral temperature gradient is not a reliable indicator of hypoperfusion. In keeping with this, a systematic review of studies between 1969 and 2009 showed a lack of consensus on the usefulness of skin temperature as a marker of hypoperfusion; the review concluded that skin temperature is not a reliable indicator of hypoperfusion when used in isolation [11].

A raised core-peripheral temperature differential is typical in the initial hours following a major burn injury [12]. The increased gap represents the body’s attempts to maintain core temperature when faced with an insult to the normal physiological thermoregulatory mechanisms and burns shock contributed to by hypovolaemia, cardiac depression and increased peripheral vascular resistance. The core-peripheral gradient has been used as a non-invasive marker of hypovolaemia and a guide for fluid resuscitation in adults [13] and paediatric burn patients [14]. Unlike other markers of hypoperfusion, including elevated base deficit [15], increased lactate [16] and reduced urine output [17], the significance of core-peripheral temperature gradients as a predictor of prognosis has not been investigated to our knowledge.

## 2. Materials and Methods

### 2.1. Aim of the Study

The difference (the gap) between the core and peripheral temperature is perceived as an integral part of patient temperature management decision making in many critical conditions. However, the timing and determinants of the gap in severe burn injuries are poorly understood. Therefore, we set out to explore the timing, magnitude, and prognostic factors of the core-peripheral temperature gap in burn injury patients.

### 2.2. Methodology

Data were collected from 116 patients admitted with severe burns to the Intensive Care Unit (ICU) at Queen Elizabeth Hospital Birmingham, University Hospitals Birmingham NHS Foundation Trust between 2016 and 2022.

Patients who had sustained a total body surface area (TBSA) burn injury of greater than 15% and were admitted to the ICU for at least 48 h were included in the analysis. Patients admitted to the ICU for palliation, or those who did not have concurrent core and peripheral temperature measurement recordings were excluded. One patient’s observations were further excluded due to physiologically implausible results, attributed to a technical error in the measurement. Sixty-one patients met these criteria, and their data, recorded on the hospital’s electronic system (PICS—Prescribing Information and Communication System, Birmingham, UK), were retrospectively analysed (Figure 1).

The study was reviewed and approved through the Clinical Audit Registration System (University Hospital Birmingham NHS Foundation Trust, registration number CARMS-14835, registered on 20 November 2018).

### 2.3. Statistical Analysis

Baseline temperature readings were defined as the first reading after admission to the ICU. Baseline hypothermia was defined as a core temperature of <36 °C, as per ISBI guidelines [1]. Target core temperature was defined as a temperature between 37.5 °C and 39.5 °C. Continuous variables were categorised as elderly patients (>65 years), high TSBA (>50), and high Baux (>100) and were assessed and recorded within 24 h of admission to ICU. In the first 24 h, inhalation injury was assessed and diagnosed via direct bronchoscopy. Core temperature was measured using an indwelled bladder catheter probe. The peripheral temperature was measured using a tympanic thermometer. External transfer was an emergency or urgent transfer from a peripheral hospital to the burns centre for the escalation of care.

Data collected included patient sex, age, presence or absence of inhalational injury, TBSA of injury, revised Baux score, hourly core temperatures, and peripheral temperatures during the first 48 h of ICU admission.

Descriptive statistics were presented using means and standard deviation (SD) for continuous variables and count and percent (%) for categorical variables. The association between baseline clinical and demographic characteristics and core-peripheral temperature gap >2 °C on admission to ICU, at 48 h, and in-ICU mortality was assessed using univariable logistic regression. Statistical significance was assumed at the 0.05 level. All analyses were conducted using Stata 17.0 or 18.5 (StataCorp LLC, College Station, TX, USA).

## 3. Results

### 3.1. Baseline Characteristics of Cohort

Across the six-year period, a total of 116 patients were admitted to the ICU burns service. Patients admitted to the ICU required either organ support or aggressive fluid resuscitation. Seventy-five patients met the inclusion criteria, and fourteen were excluded according to the pre-defined exclusion criteria, leaving sixty-one patients to be studied. The selection process details are presented in Section 2.2 (Figure 1). The baseline characteristics of the studied patients are presented in Table 1.

### 3.2. Temperature Values in the Initial 48 h

Hourly core and peripheral temperature readings were collected for each patient in the ICU during the first 48 h of admission (resuscitation period). On average, 35 (SD ± 9.9) observations were made for each patient during the first 48 h, with patients being under observation for 41.9 (±9.7) h on average. Some data were missing due to the patient being outside the ICU for operative or diagnostic procedures. The initial surgery was performed in all cases in the first 48 h. The average core body temperature in the cohort increased from 36.6 °C (±1.4) on admission to the ICU (baseline) to 37.7 °C (±0.7) at 48 h. The mean core-peripheral temperature gap at baseline was 1.4 °C (±1.7) and reached 3.6 °C (±1.6) at the end of 48 h. This change in both parameters showed a moderate positive correlation between the core temperature and the core-peripheral temperature gap at 48 h (r = 0.5, *p* < 0.001). A total of 18 patients (29.5%) were hypothermic (<36 °C), and 19 patients (31.1%) had a core-peripheral temperature gap of >2 °C at admission to the ICU. Those numbers were reduced to two patients remaining hypothermic (3.3%) and twelve patients having a core-peripheral temperature gap of >2 °C at the end of the 48 h resuscitation period (Table 2 and Table 3).

### 3.3. Predictors of Core-Peripheral Temperature Gap at the Beginning and the End of the Resuscitation Period

#### 3.3.1. Core-Peripheral Gap at Admission to ICU

Further statistical analysis was performed to analyse the association between the patient clinical and demographic characteristics at admission to the ICU and the core-peripheral temperature gap (Table 4). None of the characteristics evaluated at admission to the ICU significantly increased the risk of a core-peripheral gap above 2 °C.

#### 3.3.2. Core-Peripheral Gap at 48 h

A similar analysis was performed at 48 h after patient admission to the ICU to identify variables increasing the odds of the core-peripheral gap being above 2 °C at the end of the resuscitation period (Table 5). A higher core temperature at 48 h was significantly associated with the risk of a core-peripheral temperature gap above 2 °C. No other variables were associated with a significant risk of an elevated gap beyond 2 °C.

#### 3.3.3. In-ICU Mortality and Core-Peripheral Gap

Further analysis was performed to assess the impact of the different patient characteristics on patients’ in-ICU mortality (Table 6). There were no deaths among patients transferred from external hospitals. Two variables showed significant reductions in the odds of in-ICU mortality: a target core temperature at the end of the resuscitation period and a core-peripheral gap > 2 °C at the end of the same period. Both variables, as mentioned before, moderately correlated with each other.

Nine patients died whilst in ICU; all of them had a core-peripheral gap < 2 °C at 48 h. It is worth noting that the survivors had a higher mean 48 h gap (1.6 [95%CI: 1.3 to 1.9] compared to non-survivors 0.8 [0.2 to 1.4]; *p* = 0.04) (Figure 2).

## 4. Discussion

The current literature on temperature control in patients with major burn injury is predominantly focused on the maintenance of core body temperature and the avoidance of both hypothermia and hyperthermia, which are associated with negative outcomes [2,18,19]. Hypothermia is associated with coagulation abnormalities, cardiorespiratory complications and increased susceptibility to infection [2,20]. Hyperthermia impairs normal cellular function and occurs about 48 h after injury and is related to hypermetabolic response [21]. A previous study of ours focussed on core temperature during the entire ICU stay following burn injury and showed that hypothermia was more likely in the first 48 h, likely in keeping with initial reduced metabolic rate, iatrogenic causes including fluid resuscitation, exposed burn wounds and prior to thermoregulation interventions [22]. Non-survivors were more prone to hypothermia during their stay in ICU [22].

However, there is limited literature on the value of the core-peripheral temperature gap in managing major burns. Therefore, different guidelines discussing the gap have been based on expert opinion and evidence for the core-peripheral temperature gap in other conditions, such as septic shock. The institutional guidelines at the burns centre where this study was conducted recommend a core-peripheral temperature gap of ≤2 °C and a target body temperature between 37.5 °C and 39.5 °C during admission following a severe burn injury. The guidelines assume that a core-peripheral temperature gap above two degrees within the first 48 h of ICU admission is associated with a higher mortality rate. This was not reflected in the data from our study.

The cohort of patients who survived was found to have a higher core-peripheral temperature gap at 48 h than those who died. Furthermore, all patients in the cohort who died after 48 h had a core-peripheral temperature gap of <2 °C (Figure 2). These findings differed from the initial expert consensus and local guidelines. The reason for such a discrepancy may be multifactorial.

Previous studies looking into the core-peripheral temperature gap and its use as a prognostic factor in critically ill patients have considered patients with pathologies other than burn injuries. The prospective study by Amson and colleagues evaluating the value of the core-skin temperature gradient as a marker of early mortality in patients with septic shock had a very different result; they found that a gap of >7 °C was predictive of early mortality [9]. The differing pathophysiology of septic shock and severe burn injuries means that although there is evidence to show that a raised core-peripheral temperature gap is indicative of a poorer outcome in patients with septic shock, it is not generalisable to patients with severe burn injuries. Ambient temperature manipulation of the patient’s environment to avoid hypothermia forms an essential part of managing critically ill patients with burn injuries, along with other therapeutic interventions [23]. It is unclear what value measuring the core-peripheral temperature gap has in an environment where the patient’s temperature is actively manipulated.

Our study showed a highly varied distribution of 48 h temperature gaps in patients who survived (Figure 2). Additionally, our study had a relatively small cohort of patients because data from our unit alone were analysed. Given our study’s limitations, one cannot definitively conclude from our data alone that a lower core-peripheral temperature gap at 48 h is associated with increased mortality in patients with major burn injuries. Our data show that the factor that reduces mortality is the avoidance of hypothermia and, even more importantly, achieving a higher core temperature during the resuscitation phase. Therefore, one may hypothesise that the core-peripheral gap in patients, in whom core temperature is actively manipulated, simply reflects the ability of patients to respond to therapeutic interventions or to mount a hypermetabolic response. Our results suggest that the utmost priority should be achieving an adequate target core temperature in the initial resuscitation phase, rather than placing undue importance on the core-peripheral gap when making clinical therapeutic decisions.

Further studies, in the form of a multicentre trial, would provide a significantly larger cohort of patients and include data on temperature management in the prehospital and emergency department settings. This would allow for the stratification of the core-peripheral temperature gap and further determination of its impact on outcomes in critically ill patients with severe burn injuries.

## 5. Conclusions

A core-peripheral temperature gap above 2 °C is not indicative of poorer outcomes in adult patients with severe burns during the resuscitation period. On the contrary, in our cohort, a temperature gap below 2 °C as a single independent variable was associated with higher mortality. Mortality in those patients was also increased by hypothermia (a temperature below 36 °C) at admission and at the end of the study period. Achieving a higher core body temperature at the end of the resuscitation period had an even more pronounced, albeit opposite effect, significantly reducing mortality.

Our findings support previous studies suggesting that avoiding hypothermia and achieving a higher core temperature are associated with reduced mortality. However, it also challenges the previous expert consensus that a lower core-peripheral gap indicates poorer outcomes. Further research with a larger cohort of patients is required to identify whether a higher core-peripheral temperature gap predicts patient outcomes in critically ill patients with severe burns.

## Figures and Tables

**Figure 1 ebj-06-00011-f001:**
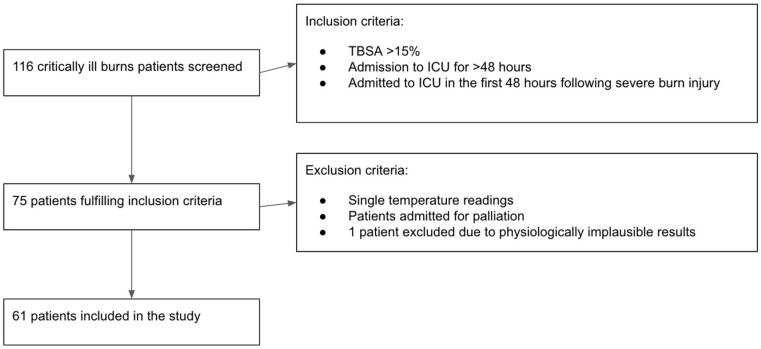
Flow chart summarising the selection of patients using inclusion and exclusion criteria.

**Figure 2 ebj-06-00011-f002:**
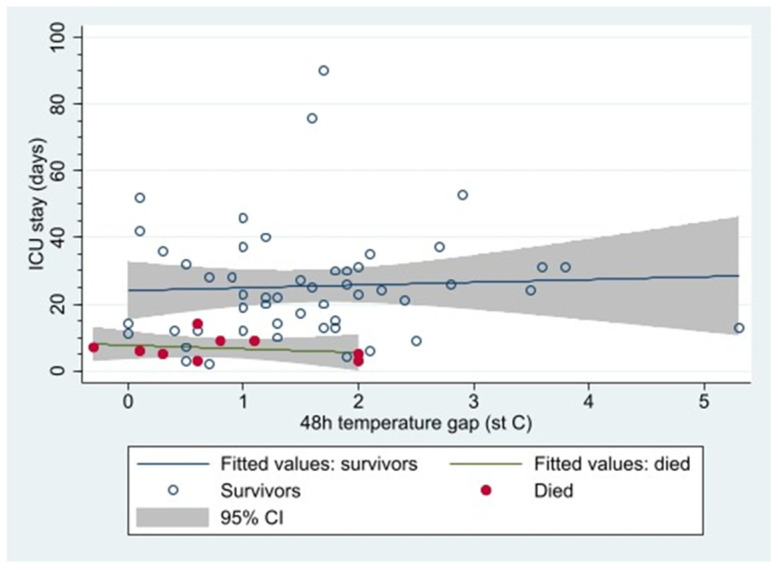
The core-peripheral temperature gap at 48 h in survivors vs. patients who died in ICU.

**Table 1 ebj-06-00011-t001:** Baseline characteristics of 61 patients studied.

	Mean	SD
Sex: Male/Female	45/16 (73.7%/26.2%)	
Age (years)	43.1	16.6
Inhalational burns (*n*, %)	37 (60.6%)	
Revised Baux score	99.2	27.4
High Baux > 100 (*n*, %)	31 (50.8%)	
TBSA (%)	45.8	20.3
High TSBA > 50% (*n*, %)	25 (41%)	

**Table 2 ebj-06-00011-t002:** Temperature values at baseline (first readings after admission to ICU).

	Mean	SD
Core temperature (°C)	36.6	1.4
Peripheral temperature (°C)	35.2	2.2
Core-peripheral gap (°C)	1.4	1.7
Hypothermia (*n*, %)	18 (29.5%)	
Core-peripheral gap > 2 °C (*n*, %)	19 (31.1%)	
Core-peripheral gap < 0 °C (*n*, %)	11 (18.0%)	

**Table 3 ebj-06-00011-t003:** Temperature values at 48 h (at the end of the resuscitation period).

	Mean	SD
Core temperature (°C)	37.8	0.79
Hypothermia (*n*, %)	2 (3.3%)	
Core-peripheral gap > 2 °C (*n*, %)	12 (19.7%)	
Core-peripheral gap < 0 °C (*n*, %)	1 (1.6%)	

**Table 4 ebj-06-00011-t004:** Determinants of core-peripheral temperature gap >2 °C on admission to ICU.

	Odds Ratio	Lower 95% CI	Upper 95% CI	*p*-Value
Female	3.091	0.938	10.188	0.064
Age > 65 y	0.871	0.153	4.948	0.876
TBSA	1.010	0.983	1.038	0.469
Revised Baux score	1.005	0.985	1.026	0.603
Core temperature (°C)	0.839	0.564	1.250	0.388
Hypothermia (<36 °C)	2.327	0.733	7.385	0.152
Flame burns	3.600	0.411	31.563	0.248
TBSA > 50	1.070	0.356	3.212	0.905
Baux score > 100	1.512	0.507	4.515	0.458
External transfer	0.236	0.027	2.039	0.189

**Table 5 ebj-06-00011-t005:** Determinants of core-peripheral temperature gap (>2 °C) at 48 h (logistic regression).

	Odds Ratio	Lower 95% CI	Upper 95% CI	*p*-Value
Age > 65 y	0.652	0.071	5.988	0.705
Female	0.923	0.216	3.945	0.914
TBSA	1.030	0.997	1.064	0.076
Revised Baux score	1.022	0.995	1.049	0.111
Baseline core temperature (°C)	1.047	0.657	1.670	0.846
48 h core temperature (°C)	**3.214**	**1.162**	**8.890**	**0.024**
Hypothermia (<36 °C)	1.250	0.324	4.826	0.746
Flame burns	0.698	0.122	3.983	0.685
TBSA > 50	3.765	0.989	14.329	0.052
Baux score > 100	3.682	0.888	15.274	0.073
External transfer	1.200	0.216	6.676	0.835

Results statistically significant highlighted in bold.

**Table 6 ebj-06-00011-t006:** Association between temperatures at 48 h and in-ICU mortality.

	Odds Ratio	Lower 95% CI	Upper 95% CI	*p*-Value
Age > 65 y	2.629	0.425	16.263	0.299
Female	1.462	0.319	6.698	0.625
Revised Baux score	1.033	0.999	1.068	0.056
Baseline core temperature (°C)	0.751	0.449	1.258	0.277
48 h core temperature (°C)	**0.247**	**0.080**	**0.758**	**0.015**
Baseline core-peripheral gap > 2 °C	1.020	0.671	1.550	0.926
48 h core-peripheral gap > 2 °C	**0.350**	**0.126**	**0.970**	**0.044**
Hypothermia (<36 °C)	2.114	0.495	9.027	0.312
Flame burns	1.000			
TBSA > 50	2.105	0.503	8.816	0.308
Revised Baux score > 100	2.250	0.507	9.993	0.286

Results statistically significant highlighted in bold.

## Data Availability

The data presented in this study are available upon request from the corresponding author.

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
