# Peer review of "Mind the Gap! Core-Peripheral Temperature Gradient and Its Relationship to Mortality in Major Burns"

_2673-1991, 2025, doi:10.3390/ebj6010011_

Round 1
Reviewer 1 Report
Comments and Suggestions for Authors
Thank you for the opportunity to review this paper, which I read with interest.
I have just a few questions.
I am not sure what the baseline readings are. Are these initial measurements? If so, time from burn to first measurement might yield additional information.
The hypermetabolic response to burn injury induces mild to moderate hyperthermia from ca. 48 hrs. I wonder if the observed data is related to the inability to generate sufficient metabolic activity. I would be interested to see analysis related to core temperature alone.
From figure 2 I surmise that the temperature gap at 48 hrs. does not have prognostic value in itself. It would be fair to note this in the discussion.
The article is thought-provoking and has implications for patient management.x
Author Response
Dear Reviewer,
Many thanks for your time and all the valuable suggestions. We have addressed the questions you raised below.
Thank you for the opportunity to review this paper, which I read with interest.
I have just a few questions.
I am not sure what the baseline readings are. Are these initial measurements? If so, time from burn to first measurement might yield additional information.
You are absolutely right; the baseline readings are the first reading after ICU admission. We have corrected the text to clarify. We fully agree that the time from the actual burn to the admission to the ICU would provide further valuable contextual information to the reader. Unfortunately, such data is not available to us. We have acknowledged this in the limitation section of our paper.
The hypermetabolic response to burn injury induces mild to moderate hyperthermia from ca. 48 hrs. I wonder if the observed data is related to the inability to generate sufficient metabolic activity. I would be interested to see analysis related to core temperature alone.
Again, thank you very much for your comment. We fully agree that the peak of the metabolic response and induced hyperpyrexia happen beyond the actual resuscitation period, beyond the 48-hour mark. The focus of the current paper is purely on the first 48 hours of the resuscitation period. Although we looked into core temperature during the whole ICU stay in our previous paper, “Temperature management of adult burn patients in intensive care: findings from a retrospective cohort study in a tertiary centre in the United Kingdom” in which we found that non-survivors were more prone to hypothermia during their stay in ICU. There was an association between rBaux score and post-operative temperature, with a 0.12°C decrease per 10 points increase in rBaux score (P = 0.04).
Driver J, Fielding A, Mullhi R, Chipp E, Torlinski T. Temperature management of adult burn patients in intensive care: findings from a retrospective cohort study in a tertiary centre in the United Kingdom. Anaesthesiol Intensive Ther. 2022;54(3):226-233. doi: 10.5114/ait.2022.119131. PMID: 36189905; PMCID: PMC10156506.
We have changed the manuscript accordingly to refer to the findings of our paper and provided an appropriate citation.
From figure 2 I surmise that the temperature gap at 48 hrs. does not have prognostic value in itself. It would be fair to note this in the discussion.
Thank you very much for your comment. We have annotated the discussion accordingly.
The article is thought-provoking and has implications for patient management.
Once again, many thanks for your valuable input and comments
Reviewer 2 Report
Comments and Suggestions for Authors
Reviewer comments
Introduction
1. To avoid any confusion with the readers, you should define core-peripheral temperature gradient and why, in general, there is a gradient, what are normal values etc.
2. R64. Is the temperature gradient really 7 degrees here, or is this a typo? That would mean temperatures, for example 35 and 42 degrees!
Materials and Methods
3. What are the indications for ICU admission in your hospital? Certain TBSA, need for intubation, need for aggressive fluid resuscitation?
4. Do you think that patients with larger burns should have a higher temperature value than <36 degrees to be labelled as hypothermic?
5. How was inhalation injury diagnosed?
Results
6. Table 1. high TBSA >50% should be High TBSA >50%
7. R147. Does “baseline” mean on admission or something else? Please clarify.
8. Did any of these patients go to the operating room during the first 48h? if so, what happened to the temperature gradient to these patients?
9. Did bigger burns have a larger temperature gap than smaller burns?
10. How do you explain the fact that survivors had a larger temperature gap than patients who died?
Conclusions
11. Do you believe that better pre-hospital management of these patients regarding temperature control is warranted? This would decrease the temperature gradient between admission and 48 hours.
Author Response
Dear Reviewer, thank you very much for your time and valuable comments, which improved our paper's clarity and quality.
- To avoid any confusion with the readers, you should define core-peripheral temperature gradient and why, in general, there is a gradient, what are normal values etc.
You are absolutely right; we have clarified the introductory section of the paper to include the definition proposed by the previous authors to add clarity and educational value to this paragraph. The manuscript was changed accordingly.
- R64. Is the temperature gradient really 7 degrees here, or is this a typo? That would mean temperatures, for example 35 and 42 degrees!
It has also surprised us, but these are the actual findings of their study (results pasted below). Therefore, we have retained the 7-degree value in our text as it is the correct citation.
Results: Day-8 mortality was 16.3%. Core-to-index finger temperature gradient >7 °C was associated with day-8 mortality (OR = 18.0, [3.02-346.14], p = 0.002). This association was still significant after adjustment to the SOFA (Sequential Organ Failure Assessment) score. A model including a high SOFA score and a core-to index finger >7 °C was effective to predict day-8 mortality (c-statistic: 0.8735 [0.770-0.976]). Core-to-index finger temperature gradient was correlated with CRT, Mottling Score, and arterial lactate levels.
Amson H, Vacheron CH, Thiolliere F, Piriou V, Magnin M, Allaouchiche B. Core-to-skin temperature gradient measured by thermography predicts day-8 mortality in septic shock: A prospective observational study. J Crit Care. 2020 Dec;60:294-299. doi: 10.1016/j.jcrc.2020.08.022. Epub 2020 Sep 2. PMID: 32949897.
Materials and Methods
- What are the indications for ICU admission in your hospital? Certain TBSA, need for intubation, need for aggressive fluid resuscitation?
In strict terms, the general admission criteria to the ICU in our hospital are guided by the Guidelines of Provision of Intensive Care Services (FICM, UK), with ICU providing level 2 and level 3 care. To avoid unnecessary confusion for the international readership, the indication is defined as potentially reversible failure of at least one organ requiring support; therefore, all patients who need at least non-invasive ventilation, aggressive fluid resuscitation, and renal support are admitted to ICU. There is no predefined TBSA per se other than the need for fluid resuscitation, necessitating ICU admission, which is a joint decision of the therapeutic team. The wording of the manuscript was changed accordingly to reflect the above.
- Do you think that patients with larger burns should have a higher temperature value than <36 degrees to be labelled as hypothermic?
It is not an unreasonable suggestion, although the actual International Society for Burn Injury ISBI guidelines define hypothermia as body temperature below 36C. Therefore, we have left the hypothermia definition per ISBI to remain in line with the international guidelines. Per your suggestion, we refer to the hypothermia definition in ISBI guidelines in the materials and methods section to avoid ambiguity.
- How was inhalation injury diagnosed?
The inhalational injury was diagnosed by direct bronchoscopy—the appropriate correction to clarify that in the text was added.
Results
- Table 1. high TBSA >50% should be High TBSA >50%
Thank you very much for spotting this typo. We have corrected it accordingly
- R147. Does “baseline” mean on admission or something else? Please clarify.
You are absolutely right; the baseline readings are the first reading after ICU admission. We have corrected it in the text to make it clear.
- Did any of these patients go to the operating room during the first 48h? if so, what happened to the temperature gradient to these patients?
It is a very valid question, and sorry for the potential confusion. Our standard practice is to have the initial surgery in the operating room within 48 hours of admission for all burn patients. Therefore, there was only one group in our study. We have added a sentence to reflect our therapeutic approach and to avoid unnecessary ambiguity.
- Did bigger burns have a larger temperature gap than smaller burns?
That is another excellent remark. Table 5 presents the determinants of increased core-peripheral gap at 48 hours. P value for TBSA is above 0.05; therefore, the only conclusion we can draw is that bigger burns do not increase the risk of the gap. Therefore, we have left the wording of the document as previously.
NO ACTION POINT
- How do you explain the fact that survivors had a larger temperature gap than patients who died?
That is a very difficult question. One of the potential explanations could be that there is an ability to mount a hypermetabolic or therapeutic response in the survivors and a lack thereof in the patients who died. Based on the data we have, it would be highly speculative. Testing such a hypothesis would require a much bigger, multicentre study. Such a need is reflected in the discussion's closing statement.
Conclusions
- Do you believe that better pre-hospital management of these patients regarding temperature control is warranted? This would decrease the temperature gradient between admission and 48 hours.
You are absolutely right. Multiple previous papers show a clear association between hypothermia on arrival to the Emergency Department and burns mortality. One can only hypothesise that it may influence the gap, but there is no evidence to support this statement yet. It would make an excellent project for the future.
Once again many thanks for your time and valuable input.
Round 2
Reviewer 1 Report
Comments and Suggestions for Authors
Thank you for your response. The changes are good. You were clearly reticent in omitting reference to previous relevant work, which further validates the article.
Reviewer 2 Report
Comments and Suggestions for Authors
The authors have addressed my concerns/questions adequately.